# Calorific Characteristics of Larch (*Larix decidua*) and Oak (*Quercus robur*) Pellets Realized from Native and Torrefied Sawdust †

**Aurel Lunguleasa *** 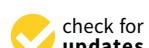**, Cosmin Spirchez and Alin M. Olarescu**

Wood Processing and Design of Wooden Product Department, Transilvania University of Brasov, 29 Street Eroilor, 500038 Brasov, Romania; cosmin.spirchez@unitbv.ro (C.S.); a.olarescu@unitbv.ro (A.M.O.)

\* Correspondence: lunga@unitbv.ro

† This paper is partially based on the conference paper: Lunguleasa, A.; Spirchez, C. Energetic aspects of oak and larch pellets obtained from sawdust waste improved by torrefaction. In Proceedings of the 8th International Conference on Sustainable Solid Waste Management, Thessaloniki, Greece, 23–26 June 2021; pp. 1−8. http://uest.ntua.gr/thessaloniki2021/pdfs/THESSALONIKI_2021_Lunguleasa_Sprirchez.pdf.

**Abstract:** This research aimed to evaluate the calorific characteristics of two biomasses from larch and oak sawdust in the form of native or torrefied pellets. Some calorific features of these two kinds of biomasses, such as ash content, higher and lower calorific values, calorific density and many others were highlighted, allowing for a comparison between oak and larch torrefied/not torrefied pellets. Installations and methods used for the process of torrefaction and for highlighting some of the calorific features were also evaluated. As a result of experiments, it was demonstrated that the larch and oak pellets were different in terms of density, but that after thermal treatment, the calorific values of both increased considerably. The investigations evidenced some increases in calorific value, up to 15.8%, for both the larch and oak sawdust/pellets. One of the main conclusions of this research was that, even though the role of biomass has diminished considerably in the last few decades, its role as a sustainable fuel remains relevant. Its use will become more widespread when the world's population understands that fossil fuels are depletable and that they must be replaced by renewable fuels such as biomass.

**Keywords:** biomass; calorific density; calorific value; larch; oak; torrefied pellet

## 1. Introduction

Generally speaking, the calorific value of fuels differs significantly from one fuel type to another. These differences are usually measurable between gaseous and solid fuels. However, solid and liquid fuels have the same rules when it comes to measuring calorific value and there are no quantifiable differences between them [1–3]. Heat in the form of hot air, obtained by heating the cold air in a given room or heating a plateaux of some kind, can be used in the food preparation or manufacturing industries for warming up the press plateau, etc. $CO_2$ from the air is captured inside the wooden biomass of trees during the growth process (about 90–100 years), forming a closed circuit due to the amount of $CO_2$ that was absorbed by the trees during the growth process being equal with that which is obtained during complete combustion (some minutes or hours). In other words, it would be accurate to say that the same amount of $CO_2$ is obtained when wooden waste/biomass is decomposed in an open space over a few years. Therefore, the use of woody waste/biomass as energy for human needs is recommended [4,5]. Natural gases have obtained a privileged position as fuels because they burn more cleanly than oil or coal, namely because that the process produces a low amount of pollution in the air. More accurately, for the same amount of fuel that is burned, natural gases spread into the atmosphere half the toxicity of coal. Also, the burning of natural gases does not release as much sulphur dioxide or

nitrates [6–8] than coals. However, one of the main problems in the field of fuels is that the natural gases are not renewable and their reserves will be depleted in the next years [9]. That is why, nowadays, thenatural gases merely represent a temporary solution in the transition to more regenerative fuels.

The term biomass pertains to large renewable energy resources such as solar, wind, geo-thermal and water energy [9]. The use of this kind of fuel has been increasing every year and is expected to become the most used type of fuel in the next few years (Figure 1).

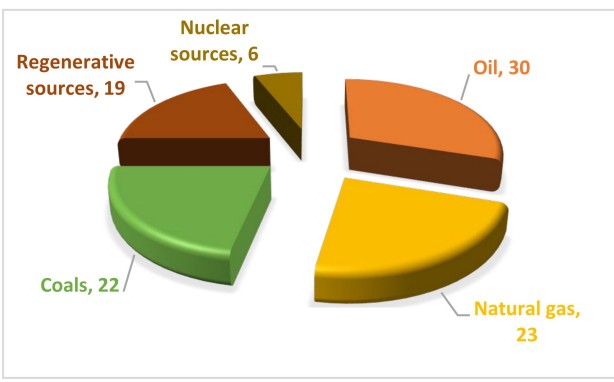

**Figure 1.** The world's energy resources [9].

Kambo and Dutta [1] highlighted some methods to improve raw sources of energy via thermal treatments. Natural waste taken from *Miscanthus* spp., pretreated by torrefaction and compacted in the form of pellets, was researched in relation to its mechanical and energy characteristics, for the purpose of improving some features. The effective density and calorific density of torrefied pellets increased from 834 kg/m$^3$ and 15.7 GJ/m$^3$ to 1036 kg/m$^3$ and 26.9 GJ/m$^3$, respectively. Akinrinola [7] showed that the thermal treatment of torrefied biomass from Nigeria (resulting from two wooden species and two other waste crops) was able to improve some fuel features, regardless of whether they were used for industrial or domestic energy sources. Other authors [10] took only waste of only two wood species, *Liriodendron tulipifera* L. and *Larix kaemferi* C., in order to make pellets for the purpose of improving resistance and calorific value by using lignin powder as an additive. Peng et al. [11] stated that good pellets can be realized from particles with small dimensions. A complete analysis of native and torrefied pellets was made by Kumar et al. [6], who identified the benefits of using wooden waste in combination with crushed coal.

Some parameters of the torrefaction/pelletizing process were made by Rudolfsson et al. [12]. This study was focused on the moisture content, particle dimensions and the extrusion channel length. Oh et al. [13] researched the pelletizing/torrefying process of sawdust waste from larch (*Larix Kaempferi* C.) and poplar (*Liriodendron Tulipifera* L.) species. Nhuchhen and Basu [14] analyzed a new methodology for replacing nitrogen, which is very expensive, with pressurized air. Meanwhile, Kudo et al. [15] replaced inert nitrogen gas, which results in the poor adhesion of coal shale in pellets, with wet saturated steam. The resistance of the pellets obtained because of this change increased by five times. Peng et al. [16] observed that the biomass should first be torrefied and then compacted by pelletizing. Thus, pellets with superior features, namely a high resistance to water, can be obtained.

Eseyin et al. [17] stated that torrefied biomass in the form of pellets/briquettes or native biomass has a high calorific energy, is resistant to water and fungus and has a longer storage time. Additionally, the advantages of using biomass include the fact that biomass is an environmentally friendly fuel that produces substantially lower $CO_2$ emissions compared to fossil fuel. Granados et al. [18] studied the influence of the torrefaction installation on the process, performed in two stages, for small wood particles of 0.5–1 mm from poplar species. During the torrefaction process, instead of an inert nitrogen atmosphere, the resulting gases were used for the heat treatment process. The mass loss was 34%

and the calorific value increased by 40% when the torrefaction temperature was 300 °C. Okoro et al. [19] studied the role of wooden biomass obtained from two species of pine versus fossil resources, especially when they were thermally treated to obtain syngas via pyro-gasification. The optimum thermal treatment was at 300 °C for 45 min. The calorific value increased by 57%. Olugbade and Ojo [20] determined a number of the energy characteristics of biomass and in particular for heat-treated agricultural biomass by torrefaction in an inert nitrogen atmosphere. Torrefaction is considered to be a process of medium pyrolysis. Alokika and Singh [21] optimized the parameters of the torrefaction process for an acacia species. TGA, SEM and FTIR analyzes showed an 18% higher calorific value, a 75% increase in fixed carbon, and an energy efficiency at 252 °C and a treatment time of 60 min, compared to raw materials. Pérez et al. [22] treated four fast-growing species from Colombia, namely Gmelina, two species of pine and one type of eucalyptus, respectively. No major differences were found between the energetic properties of these species (a volatile content of 70%, calorific value 18.7 MJ/kg and ash content below 1%) but they behaved differently during the thermal treatment process. It has also been shown via chemical analysis that only hemicellulose was degraded during torrefying, whereas lignin and cellulose remaining undegraded. Lee et al. [23] used sewage sludge for torrefaction to enrich energetic properties. The obtained product was very similar to coal. The spontaneous ignition of this fuel has been shown to occur at a temperature of 211 °C. Further research has focused on both the use of other residues obtained from other industries, such as from the coffee [24] and paper [25] industries, or the internationally correlation of standardized properties [26].

An initial deduction that can be made from the above studies in the field of sawdust thermal torrefaction is that, although torrefaction is a very efficient treatment, the nitrogen flow used during biomass torrefying is not cheap and requires complex installation. This is the reason why some simpler torrefaction methods have been tested to achieve the same effects [14,15,18]. A second conclusion that can be made is that more and more studies refer to the use of lignocellulosic biomass as a safe and sustainable method of transitioning from fossil fuel energy to the energy obtained from renewable sources.

Objectives: The aim of this paper was to improve the calorific features of the sawdust of two species (one of the softwoods and another of the hardwoods, in order to observe their different behaviors), torrefied over 200 °C without the admission of air during the thermal process. Some calorific characteristics were tasted for the native/torrefied pellets. In particular, the influence of moisture content on the calorific value of the two analyzed species was studied.

## 2. Materials and Methods

For the experimental, two lignocellulosic biomasses in the form of native sawdust were used. Sawdust was obtained from larch and oak timber. Waste sawdust was collected from a circular saw machine at a university wood processing workshop (Transilvania University of Brasov, Brasov, Romania) when the two wood species, in the form of timber, were processed. As the sawdust purchased had large particles and spalls, it was sorted using a 6 mm x 6 mm mesh sieve. After sorting, only the fraction that fell through the sieve was taken, with the remainder that remained above the sieve (about 12%) being eliminated. Next, the moisture of the sawdust was determined, obtaining values within the limits of 10 ± 0.6% (because the dry timber was used due to the needs of the workshop). The experimental process involved three steps: a first step, in which the characteristics of the raw material were determined; a second step, in which the small sawdust was torrefied and the native/torrefied pellets were made; and a third step for testing the obtained pellets.

Granulometry. In order to determine the characteristics of the sawdust, its granulometry was determined using sieves with 4 mm × 4 mm, 3.15 mm × 3.15 mm, 2 mm × 2 mm, 1.25 mm × 1.25 mm and 0.8 mm × 0.8 mm meshes arranged on an electric device with vibration. Six sawdust samples were used for each species analyzed, and the results obtained in the form of masses were transformed into percentage values by reporting the value of

the mass remaining on the sieve to the total mass of the sample. As an example, Equation (1) shows how the percentage of chips was determined for the fraction 3.15 mm × 3.15 mm.

$$P_{3.5} = (m_{3.15} : m_s) \times 100 \ [\%] \tag{1}$$

where: $P_{3.15}$—the participation percentage of the fraction of 3.15 mm × 3.15 mm from the total chips in %; $m_{3.15}$ is the mass of the fraction of 3.15 mm × 3.15 mm which remains above the respective sieve in g; and $m_s$ is mass of the whole sample in g.

Similar to Equation (1), the other five percentage values were determined for sieves with meshes of 4 mm × 4 mm, 2.5 mm × 2.5 mm, 1.25 mm × 1.25 mm, 0.8 mm × 0.8 mm, 0.4 mm × 0.4 mm and the rest (which remained below the sieve of 0.4 mm × 0.4 mm and was collected in the existing collector cylinder). The values for all six groups of tests were then averaged, and, based on these, the graphs of variation of the obtained values were realized.

Bulk sawdust features [26]. In the case of sawdust, its bulk density was determined by using a graduated cylinder to determine the volume of sawdust, the calculation relation being as follows Equation (2):

$$\varrho = (m_t - m_c) : (\pi \times d^2 \times h) \times 10^6 \ [\mathrm{kg/m^3}] \tag{2}$$

where: $m_t$ is the total mass of the sawdust with the cylinder in g; $m_c$ is mass of the empty cylinder in g; d is inner diameter of the cylinder in mm; and h is height of the sawdust layer inside the cylinder in mm.

This bulk sawdust density was used to determine the expanding or compression coefficient of the sawdust used, using the following two equations, Equation (3):

$$Ke = \varrho w : \varrho s; \ Kc = \varrho ws : \varrho s \tag{3}$$

where: Ke is the expanding coefficient; Kc is the compression coefficient; $\varrho_w$ is density of wood in $\mathrm{kg/m^3}$; and $\varrho_s$ is the bulk density of sawdust in $\mathrm{kg/m^3}$.

The density of the solid wood was determined by taking 10 specimens with dimensions of 20 mm × 20 mm × 30 mm from the remains of the timber pieces, with the mass in lreation to their volume being reported. This Ke coefficient was used to determine the compressibility of the sawdust, comparing the density of the solid wood from which it came, with this of the sawdust with respect to its dimensional peculiarities.

Obtaining pellets from native sawdust. Pellets with a diameter of 10 mm were obtained from the sorted sawdust, using a laboratory pelletizing device (as part of calorimeter apparatus). The main characteristics that were determined for these native pellets were size, moisture content, unit density, bulk density, higher and lower calorific value (HCV, LCV), calorific efficiency, combustion time and calorific density, etc.

Ash content determination. A well-known and standardized method was used to determine the ash content of the sawdust (ASTM E1755-01) [25]. The wood sawdust was dried in a laboratory oven at 105 °C for 2 h in order to eliminate the influence of moisture content on the ash content. For this purpose, metal crucibles made of nickel–chromium alloys that are resistant to high temperatures were used. These were cleaned thoroughly and then burned on a butane gas flame, before being cooled in a desiccator and weighed up to three decimal places on a Kern, Germany electronic scale. A small amount of less than 1 g of sawdust was then placed on the surface of the crucible in 2–3 layers and then the crucibles were weighed again. In order to protect the inner cavity of the calciner (Protherm, Ploiesti, Romania), the sawdust crucible was burned on a flame of butane gas until there was no more flame and smoke, and the ash obtained had a black color. At this point, the crucible with the ashes was weighed again with the help of the high-precision balance up to 3 decimal. Next, the crucibles were introduced into the calcination furnace for a period of about 30 min at a temperature of 650 °C. After that, in order to determine if the calcination process was finished, the crucibles were checked to see if they had a light greyish ash

color and that there was no longer any spark and carbon. When the ash content was determined, the sawdust samples were completely dried out in an oven and the crucible mass was also kept into consideration, as seen in Equation (8) [19–21]. The black ash and the calcined ash masses were determined as proportions with the following relationships using Equations (4) and (5):

$$\text{BAc} = (m_{ba+c} - m_c) : (m_{s+c} - m_c) \times 100 \ [\%] \tag{4}$$

$$\text{CAc} = (m_{ca+c} - m_c) : (m_{s+c} - m_c) \times 100 \ [\%] \tag{5}$$

where: $m_{ba+c}$ is mass of the black ash and the crucible in g; $m_{ca+c}$ is the mass of the calcinated ash and the crucible in g; $m_{s+c}$ is the mass of specimen and the crucible in g; and $m_c$ is the empty crucible mass in g.

Torrefaction of sawdust. Lignocellulosic biomass in the form of larch and oak sawdust, taken from a circular saw and sorted, was subjected to a torrefaction process inside of the calcination furnace STC 18.26 (Ploiesti, Romania) without air admission. Different temperatures, of 200, 220, 240, 260, 280 and 300 °C, and times, of 3, 5 and 10 min, were used to highlight its calorific properties. Each type of torrefying regime was performed individually (not progressively, from a lower treatment to a higher one), obtaining a total of 18 types of regimes for larch and another 18 for oak sawdust. For each regimen, 10 individual and distinct samples were used. The heat treatment was performed without oxygen intake. The torrefied samples were cooled and subjected to specific tests, mainly to determine mass loss. This material was then transformed into torrefied pellets so that they had the same dimensional characteristics as the native ones (untreated). The torrefaction process was performed on a small amount of wooden sawdust for each temperature and time [22–26]. During this period, some changes in the sawdust color occurred [27–29]. The mass loss percentage (ML) of the torrefied sawdust was determined with using Equation (6):

$$\text{ML} = (m_i - m_f) : m_i \times 100 \ [\%] \tag{6}$$

where: $m_i$ is the initial sawdust mass, before torrefaction, in g and $m_f$ is the final sawdust mass, after torrefaction, in g.

The ML values represented the mean of 10 tests, each of them for all periods, temperature, and wooden species.

Density of pellets. The densities of the native and treated pellets were obtained as the ratio between their mass and volume when they had the same Mc, i.e., 10%. Because the form of the pellets had been approximated as a right cylinder (their heads were polished in order to for them to have a perpendicular to length cross section), the relationship of density was determined by Equation (7):

$$\varrho = 4 \times m : (\pi \times d^2 \times l) \times 10^6 \ [\text{kg}/\text{m}^3] \tag{7}$$

where: m is the pellet's mass in g; d is the pellet's diameter in mm; and l is the pellet's length in mm.

High and low calorific values. The apparatus that was used to determine the calorific energy of the solid biomass in the form of sawdust pellets was the calorimeter bomb, type XRY-1C, offered by Shanghai Changji Trading Company Limited., China. The calibration of the calorimetric apparatus was performed before testing by using benzoic acid that had a calorific value of 26,463 kJ/kg. By using this value, the coefficient k was obtained with the same method of determination and the same CV relationship using Equation (8).

$$\text{CV} = (k \times (T_f - T_i) - qi) : m \ [\text{kJ}/\text{kg}] \tag{8}$$

where: CV is the best calorific value for 0% Mc in KJ/kg; k is the calorimeter coefficient in kJ/Celsius degrees; $T_f$ is the final temperature in °C; $T_i$ is the initial temperature in °C; m

is the mass of the specimen in kg; and $q_i$ is the supplementary heat given by cotton and nickel wire burning in KJ.

The calorimetric installation software for determining the calorific value provided both the high and low calorific values and the burning time of the determination. In order to find the average value of the calorific value, 8–10 valid replicates were used.

Calorific density and burning rate. The purpose of the caloric density determining was to determine the amount of energy that existed in relation with biomass volume and the necessary amount to determine the transport capacities of the truck or the storage silo of the thermal power plant. The calorific density (CD) was obtained by taking into account the calorific value of the pellets and their effective density by using Equation (9):

$$CD = CV \times \varrho \ [kJ/cm^3] \tag{9}$$

where: CV is the calorific value in kJ/kg and $\varrho$ is the density of the oven-dried pellets in $kg/m^3$.

The combustion speed of the biomass in the calorimeter was necessary in order to determine how fast the pellets burned, and in most cases this characteristic is used to determine the efficiency of the thermal power plant. Taking into account the burning time, the calorific value and the oven-dry mass, the burning rate was determined using Equation (10).

$$BR = CV: t \times m_0 \ [kJ/min] \tag{10}$$

where: BR is the burning rate in KJ/min; CV is the calorific value expressed on a dry basis in KJ/kg; $m_0$ is the mass of the oven-dried pellet in g; and t is the combustion time in min.

The mean of BR value was obtained as an arithmetic mean of eight experiments, taking in consideration different values of time and calorific value, for each sample.

Influence of moisture content on calorific value. Because the moisture content of pellets was one of the principal factors that influenced the calorific value, a relationship of dependence for 0% and other moisture content values [17] could be found by using Equation (11):

$$LCV_{Mc} = (CV \times (100 - Mc) - 2.44 \times Mc):100 \ [MJ/kg] \tag{11}$$

where: $LCV_{Mc}$ is the low calorific value at a certain moisture content in MJ/kg; CV is the calorific value for 0% moisture content in MJ/kg; and Mc is the moisture content in %.

In order to determine this influence, the larch and oak pelleted specimens were kept in a climatic chamber until three moisture content values (10%, 20% and 50%) were obtained. The high and low calorific values (HCV, LCV) were determined from these tests. Thus, two straight lines were obtained in the x0y plane (CV0Mc). The linear equations intersected the CV axis at the same point, and the horizontal Mc axis at two other different points. The arithmetic mean of the two horizontal intersections was defined as "limitative Mc". For example, if the larch pellets had a moisture content of 20%, a calorific value of 15,499 kJ/kg was obtained, and for a moisture content of 50%, a calorific power value of 10,057 kJ/kg was obtained. In this way, two points, A (50; 10057), and B (20; 15499) could be highlighted. A line can be made through the two points, whose gross equation was found using Equation (12):

$$(y - 10{,}037): (15{,}499 - 10{,}037) = (x - 20): (50 - 20) \tag{12}$$

By performing the calculations in the previous Equation (12), the final linear equation of the form $y = n - m \cdot x$ was obtained, with the calorific value being an unknown variable and the moisture content being a known variable, as seen in Equation (13):

$$HCV = 19{,}140 - 150.8 \cdot Mc \tag{13}$$

This equation is interpreted in the sense that it has a coefficient of linear equation of 19,140 and a tangent of the slope of the linear equation of $-150.8$, which means that

it has an angle of over 80 degrees, and the calorific value for completely dry pellets is 19,140 kJ/kg.

In the second part of the experiments, the torrefied sawdust was compressed into cylindrical pellets (with a dimension of 10 mm, a mass of 0.5–0.8 g, and a length of 9–11 mm) using the same press (as part of a calorimeter bomb) (Figure 2). At least 12 pellets were obtained from each native/torrefied sawdust lot. Also, two different categories of pellets, one made from larch sawdust (*Larix decidua*) and the other made from oak (*Quercus robur*) sawdust, were made and analyzed.

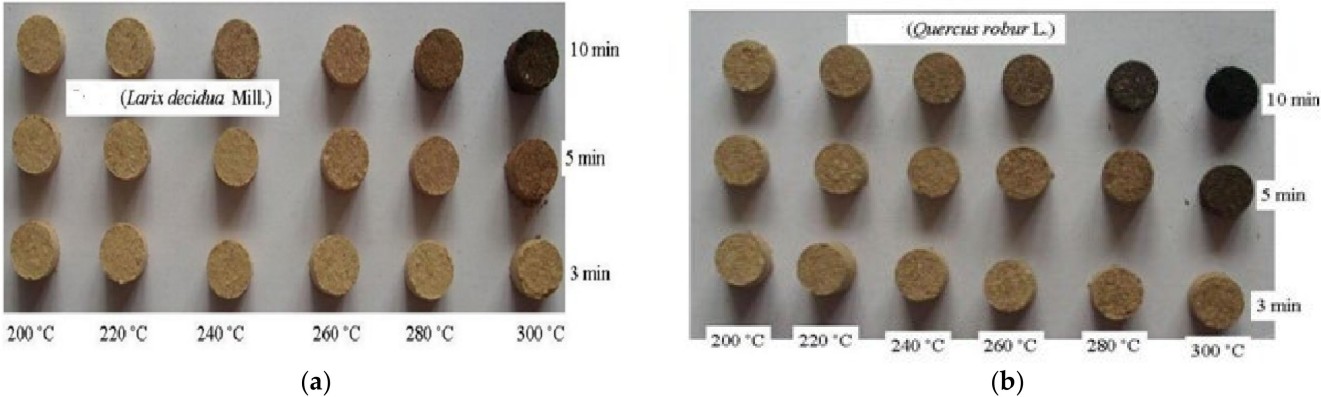

(**a**)　(**b**)

**Figure 2.** The torrefied pellets for larch (**a**) and oak (**b**).

The improvement in calorific value. The improving in calorific value after the thermal treatment of torrefaction was determined based on the calorific value obtained before and after treatment, depending on two parameters (temperature and treatment time), with aid of the next relation Equation (14):

$$I_{CV} = (Cvat - CVbt) : CVbt \times 100 \ [\%] \tag{14}$$

where: $I_{CV}$ is the increase in calorific value (CV) in %; CVat is the calorific value after torrefaction in kJ/kg; and CVbt is the calorific value before torrefaction in kJ/kg.

**Calorific efficiency**. The calorific efficiency was determined in relation to moisture content. This determination was based on the fact that an increase in moisture content implicitly leads to a decrease in calorific value, and thus it was important to determine the percentage from the maximum caloric power (for MC = 0%) that was actually used during combustion and how much was lost through the drying of the pellets. This determination was based on tests performed to determine the influence of moisture content on the calorific value by referring to the maximum obtained calorific value (CV). The calculation was performed as follows (Equation (15))

$$Cef = CV_{Mc} : CV \times 100 \ [\%] \tag{15}$$

where: $CM_{Mc}$ is the calorific value for a certain moisture content in kJ/kg and CV is the maximum calorific value for a moisture content of 0%.

Proximate analysis was performed by determining volatiles, fixed carbon and ash content (Ac). The same calciner was used by means of a heat-resistant vessel with a lid, so that the sawdust did not oxidize during the elimination of volatile matter (VM). The amount of fixed carbon (FC) that was obtainedwhen the sawdust had a 0% moisture content, was calculated using the following Equation (16):

$$FC + VM + Ac = 100 \ [\%] \tag{16}$$

**Statistical analysis**. In the first stage, the obtained values were subjected to the determination of the survey median and the standard deviation in order to observe the

trend and the scattering of the values. Using Microsoft Excel, the standard error was applied to the graphs obtained for a 95% confidence interval. The obtained trend equations were chosen in such a way that the determination coefficient $R^2$ had a value higher or was appropriated to 0.9. Analysis of variance (AVOVA one-way) was used to compare two groups of values in order to study the dependency between a dependent variable (oak density) and an independent variable (larch density). The Minitab 18 program was also used for statistical analysis in order to obtain statistical graphs and determine other statistical parameters.

## 3. Results

### 3.1. Granulometry of Native Sawdust

The granulometry of the native sawdust (Figure 3) was almost identical for larch and oak, this being determined by the fact that the raw material was taken from the same circular saw and was sorted with the same 5 mm × 5 mm sieve.

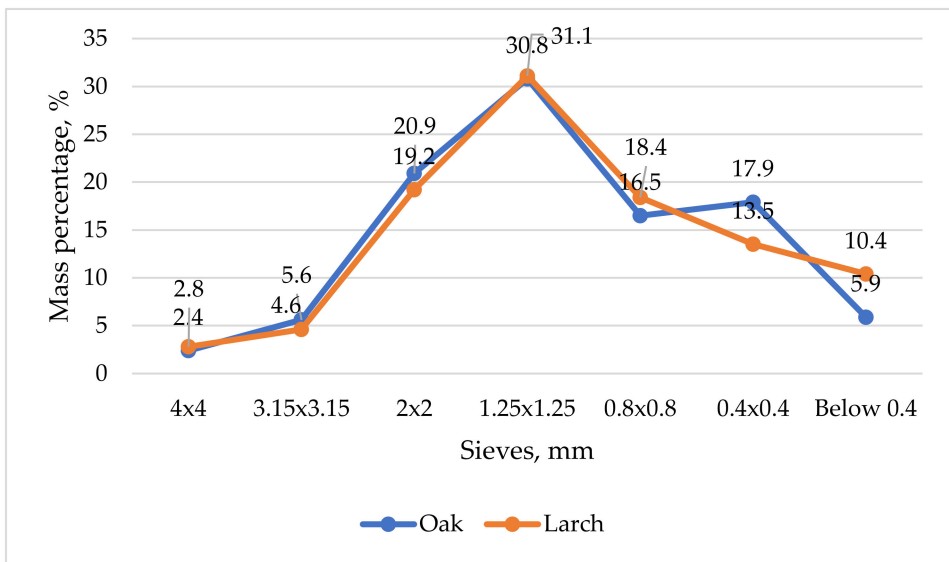

**Figure 3.** Granulometry of native sawdust.

The small variations observable in Figure 3 were determined by the different density of the two considered wood species

### 3.2. Bulk Characteristics of Sawdust

Following the procedure expressed in the previous chapter and Equation (2), a bulk density values of 162 kg/m$^3$ and 204 kg/m$^3$ were obtained for the larch and oak sawdust, respectively. The difference between the values was due to the large differences in the effective density of the two species, the calculated densities of oak and larch wood being 784 kg/m$^3$ and 575 kg/m$^3$, respectively. Moreover, the degree of compression of the sawdust of the two species was determined, taking into account the density of the wood substance (as the maximum value at which it can be compressed) for both species, which is 1450 kg/m$^3$. Using Equation (3), expansion coefficient values of 3.54 and 3.84 were obtained for larch and oak, respectively, as were other compression coefficient values, namely 8.9 for larch and 7.1 for oak. The compression ratio of larch was higher because the compaction started from a lower sawdust density value (162 kg/m$^3$) than in the case of oak (204 kg/m$^3$). Thus, the difference related the maximum value that can be reached in terms of compression was much larger. From this analysis, it can be concluded that heavier species (such as oak) have a lower compression than light species (such as larch).

### 3.3. Density of Native/Torrefied Pellets

The moisture content of the sawdust and pellets was about 10%, with this being determined by the classical gravimeter method, standardized by EN 14774-1:2009. The density of the native (un-torrefied) pellets was determined as a ratio between mass and volume and was about 1010 kg/m$^3$ for oak and 1012 kg/m$^3$ for larch. The density values were very appropriate because the dimensional characteristics of the sawdust were the same (a fraction smaller than 5 mm $\times$ 5 mm was removed from the circular sawdust) and the pelletizing press and its parameters were the same in both cases.

### 3.4. Calorific Features

For both the two analyzed species (*Quercus robur* and *Larix decidua*) the burning time, higher and lower values of calorific power (HCV and LCV), lineal equation for each type of calorific value and limitative MC were calculated, specifically when native (no-treated) pellets were used (Table 1).

**Table 1.** Calorific characteristics of native [un-treated] pellets.

| Species | Time of Burning | Calorific Value | | | Lineal Equation | Limitative Mc, % | |
|---------|-----------------|------|------|------|-----------------|------|------|
| | | HCV | LCV | CV | | | |
| Oak | 25 | 18,564 | 18,563 | 18,569 | HCV = 18,569 − 176.4·Mc | 105.2 | 100.1 |
| | | | | | LCV = 18,569 − 195.1·Mc | 95.1 | |
| Larch | 29 | 19,135 | 19,132 | 19,140 | HCV = 19,140 − 150.8·Mc | 126.9 | 97.7 |
| | | | | | LCV = 19,140 − 279.4·Mc | 68.5 | |

As a general rule, the calorific value was proportional to depended indirectly on the moisture content, with both the both higher and lower calorific values decreasing with an increase in Mc. Ideally, if the pellets under analysis were absolutely dry, i.e., they would ideally have a moisture content of 0%, then the two values, HCV and LCV, would be equal and would have the unique value CV [28]. In reality, although the pellets were dried to a constant mass (i.e., they had 0% Mc), it was seen that the higher and lower calorific values were slightly different from each other. The explanation for this is the fact that a cleaner water with a volume of 3 mL was added into the bomb in order to absorb nitrogen compounds during burning. This explains the deviation from a moisture content value of 0% (to 2–4%) and, consequently, from normal values of CV. The addition of liquid water was strongly recommended by the calorimeter provider.

### 3.5. Moisture Content Dependence on Caloric Power

Based on the methodology presented in the previous chapter, the graphs in Figure 4 were obtained, both for the native oak sawdust and for the larch sawdust. The graphs show once again that the calorific values of the two species were different, with the calorific value of larch (19,140 kJ/kg) being 3% higher than that of oak. The slight increase in calorific value for larch was due to the existence of a small amount of resin (which has a very high calorific value of about 34,000 kJ/kg) [24,26].

Figure 4 shows that with an increase of Mc, the calorific power will drop due to a certain amount of energy tat was used to dry the wooden pellets. The caloric value for 0% moisture content, noted with CV, was obtained by mathematically determining the regression equation. These lineal equations are observed in Table 1. Cuttings of the horizontal axis of each lineal equation determined the limitative moisture content. When there was no rise in caloric energy, the mean values of limited moisture content were 100.1% and 97.7%. The explanation for this is that the pellets' energy was equal to the energy used to consume water from wood (i.e., to dry out wood). When the moisture content is zero, both the HCV and LCV values were intersected, and a point with the same calorific value, CV, was obtained. This is why the calorific efficiency issue occurred during the burning of the pellets with a certain Mc [29].

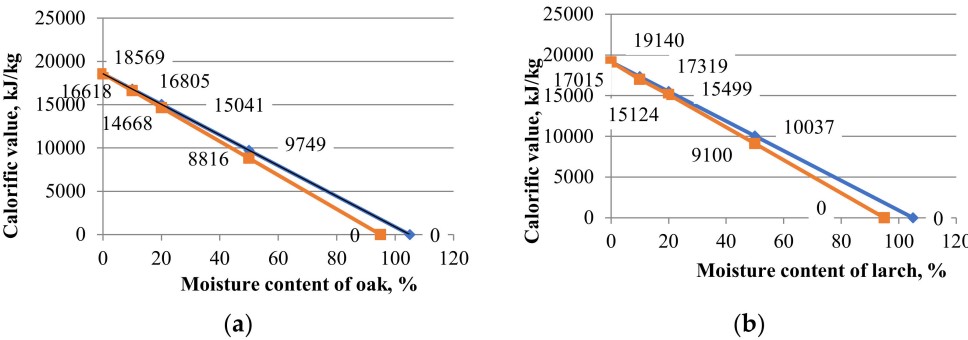

**Figure 4.** Calorific values for oak (**a**) and larch (**b**) in relation to moisture content.

### 3.6. Calorific Efficiency

With regard to calorific efficiency, Figure 5 shows that this property decreased with an increase in moisture content for oak and larch sawdust. Biomass with a 10% moisture content offered a better efficiency (95%) than that with a moisture content of 50%, the efficiency of which was 52%.

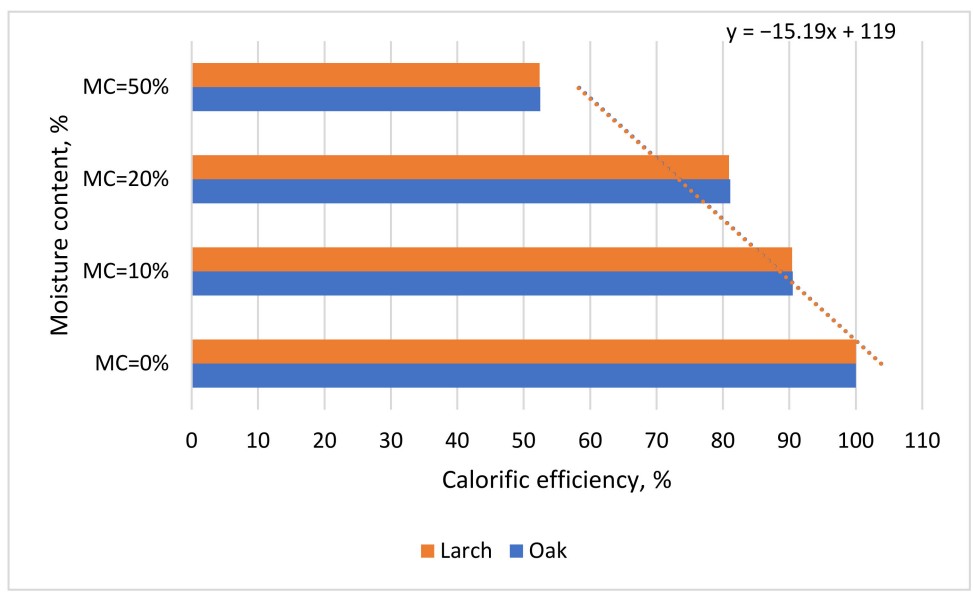

**Figure 5.** Calorific efficiency for larch and oak pellets.

### 3.7. Calorific Density

Regarding the energy properties of sawdust pellets, it was observable that the calorific density had extreme values, of 1.83–13.4 kJ/cm$^3$ for native larch pellets and 8.07–12.18 kJ/cm$^3$ for native oak pellets (Table 2). As a general rule, the calorific density decreased with an increased in pellet moisture content, with this decrease being 7.3 times for larch pellets and only of 1.5 times for oak pellets. These values show that the oak pellets were more homogeneous and had a more constant combustion. Also, the volume of the silo or means of transport would have to be slightly higher in the case of larch pellets, especially for a moisture content higher than 10%.

**Table 2.** Burning rate and calorific density.

| Features | | Moisture Content | | | |
|---|---|---|---|---|---|
| | | 0% | 10% | 20% | 50% |
| Burning rate, kJ/min | Larch | 463 | 313 | 86 | 47 |
| | Oak | 372 | 346 | 277 | 164 |
| Calorific density, kJ/cm$^3$ | Larch | 13.4 | 7.12 | 2.63 | 1.83 |
| | Oak | 12.18 | 11.06 | 10.89 | 8.07 |

### 3.8. The Burning Rate

The burning rate decreased with an increase in moisture content (Table 2). The extreme values obtained for a moisture content of 0% and 50% showed a decrease of 9.8 times for larch pellets and 2.2 times for oak pellets. It was observed with respect to burning speed, that the oak pellets were more homogeneous, that their burning would be more constant and that there would be smaller differences when the moisture content varied within the same group of pellets.

### 3.9. Mass Losses in Time of Torrefaction

Generally, it was determined that when the degree of torrefaction is increased (given by the values of time and temperature), the mass loss will also proportionally increase (Figure 6). Temperatures over 260 °C increased the mass loss [23–26] of oak pellets to a greater degree. A temperature of 300 °C represented the ideal temperature for torrefaction but was the highest one possible before self-burning occurs. Knowing that the mass loss for non-torrefied sawdust is zero, the mass loss over the total temperature range for 3 min was 8.84%, for 5 min was 16.95% and for 10 min was 40.24%. In the case of larch sawdust, the increase was of 7.62% for 3 min, 16.95% for 5 min and 17.59% for 10 min. It was observed that the oak sawdust had higher losses than the larch sawdust for all the treatment regimens and had a maximum increase of 128.7% for 10 min and 300 °C. Another conclusion was the observation that there was a substantial increase in losses when the temperature increased from 280 to 300 °C for both sawdust species, with a difference of 31.4% for oak and 9.9% for larch.

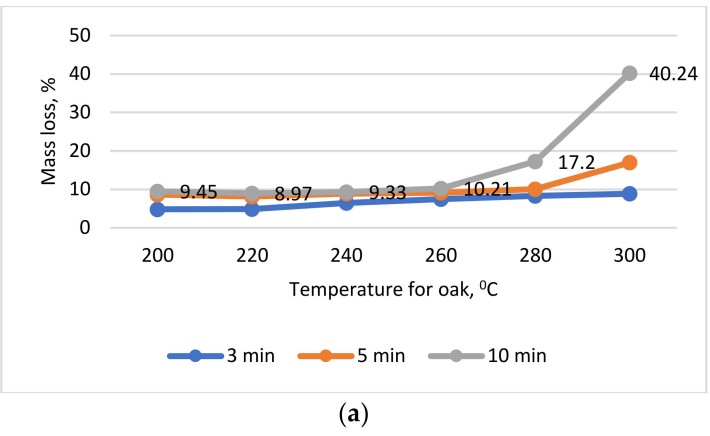

(**a**)

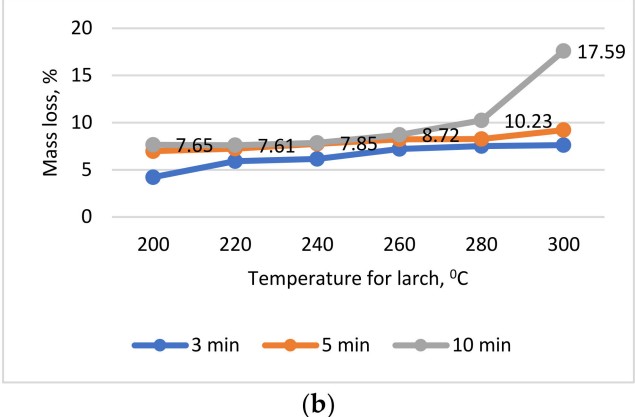

(**b**)

**Figure 6.** Mass losses for oak (*Quercus robur* L.) (**a**) and larch (*Larix decidua* Mill.) (**b**) sawdust during torrefaction.

### 3.10. Increasing the Calorific Value When the Torrefaction Process Occurs

With respect to the influence of temperature on the CV of *Quercus robur* pellets, Figure 7 reveals the increase in CV for treatments of 3 min, 5 min and 10 min in the case of larch pellets. Figure 8 reveals the increase in CV in the case of oak pellets.

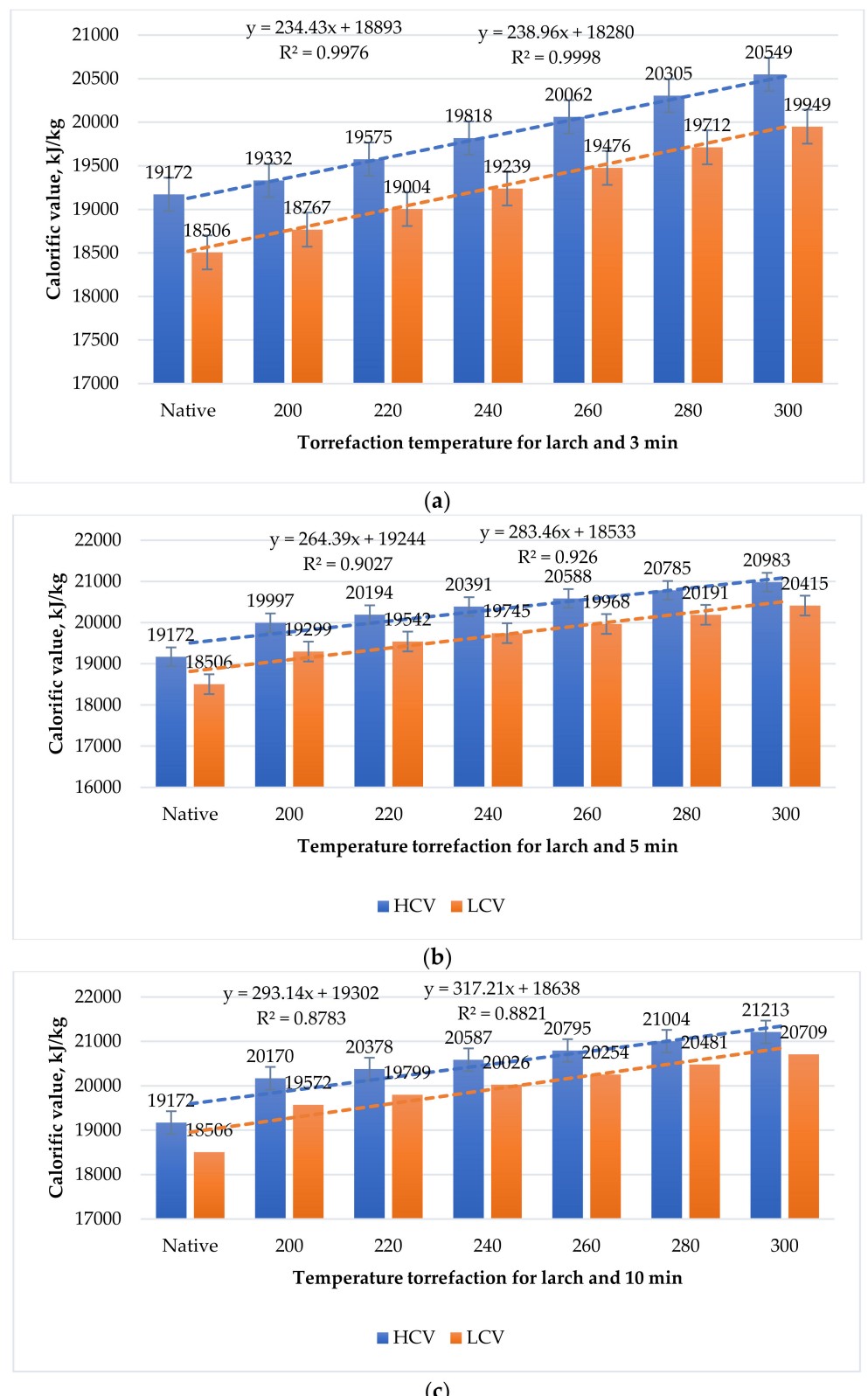

**Figure 7.** Calorific values for larch torrefaction for 3 min (**a**), 5 min (**b**) and 10 min (**c**).

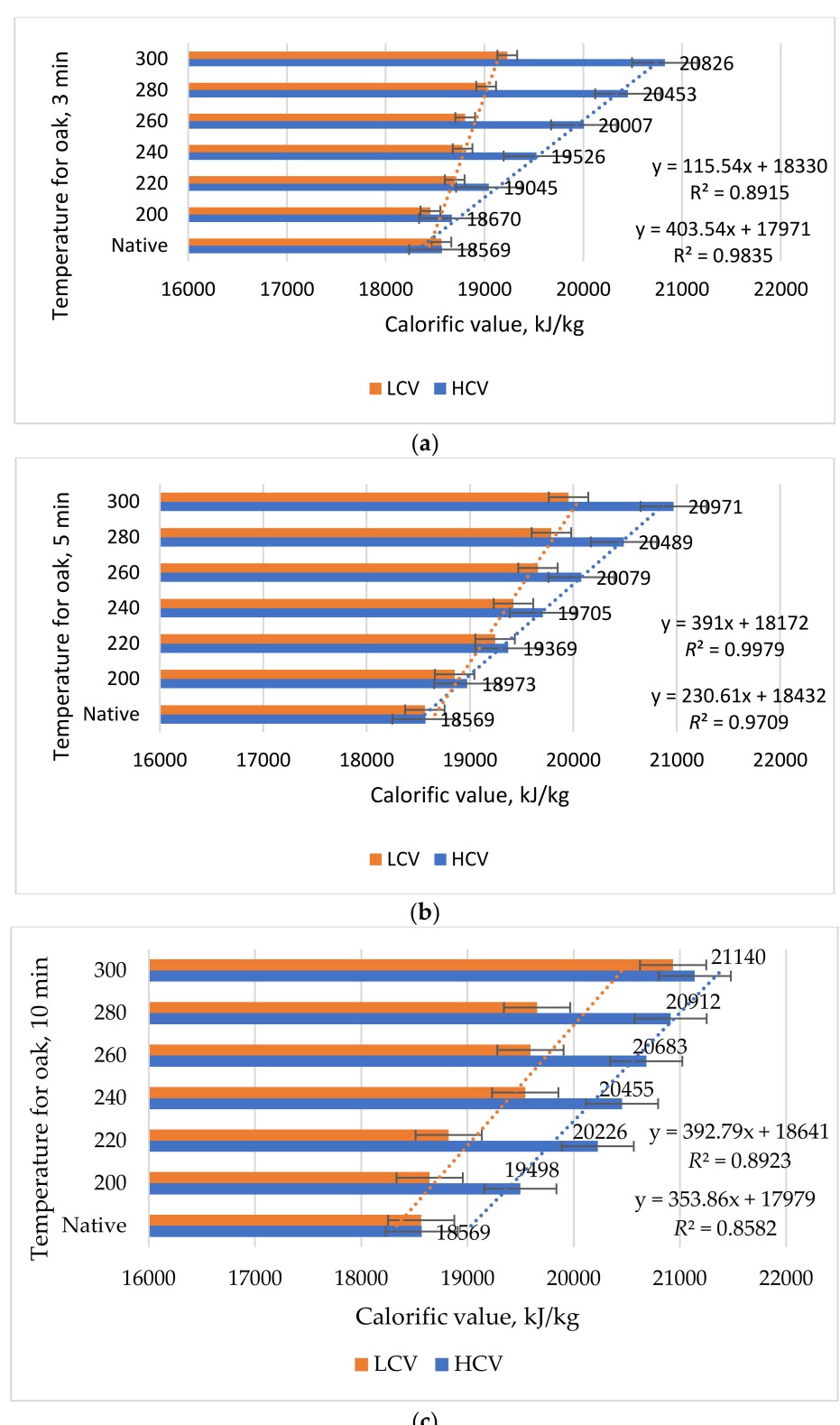

**Figure 8.** Calorific value of oak sawdust during torrefaction for 3 min (**a**), 5 min (**b**) and 10 min (**c**).

Regarding the influence of torrefaction temperature and time on the CV of the pellets obtained from the larch sawdust (Figure 7), it can state that the highest increase in CV, 15.8%, was obtained from the maximum torrefaction regime.

Throughout the heat treatment (0–300 °C and 10 min) of the larch sawdust, the HCV increased, a with maximum of 15.5%. It was observed that this increase does not correspond

to the maximum mass loss, 17.9%, because of other volatile substances that did not increase the calorific value, with the main factor influencing the increase in calorific value being the degradation of hemicelluloses [5].

As in the case of oak sawdust (Figure 8) was seen, the growth of HCV in the case of the larch does not correspond to the loss of mass, the former being much smaller. It should be noted that the 3 min period of torrefaction resulted in an increase of less than 5% in case of larch and oak sawdust. Also, the increase in calorific power to over 21,100 kJ/kg made it possible to classify the torrefied pellets as inferior coke coals.

### 3.11. Ash Content, Volatile Matter and Fixed Carbon

Different values were obtained for the ash content of native sawdust, with a value of 0.42% for the *Larix decidua* and one of 0.51% for the *Quercus robur* sawdust (Figure 9). The ash content increased slightly as a result of torrefaction. On the other hand, the sawdust that was torrefied had a high ash content, but this was not totally in concordance with the mass losses found during torrefaction [26], which is usually lower. For instance, the higher values of ash content for torrefied sawdust highlighted during laboratory tests at 300 °C was 0.58% (an increase of 13.7%, whereas the mass loss was 40.2%) for oak and 0.47% (an increase of 11.9%, whereas the maximum mass loss was 17.5%) for larch. As other authors have stated before [5,13], the main constituents of ash are silicates, oxides and hydroxides, sulphates, phosphates, carbonates, chlorides and nitrates, which means that it can be used as a fertilizer in agriculture.

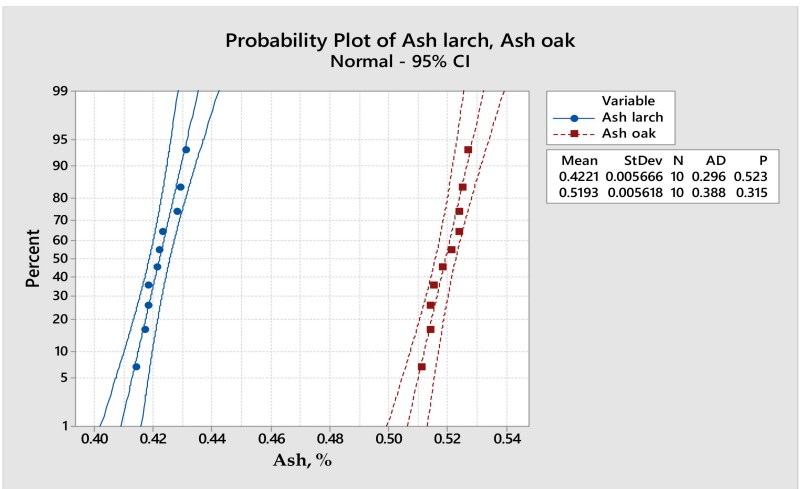

**Figure 9.** Probability plot for larch and oak ash content.

Proximate analysis of native and torrefied sawdust is available in Table 3.

**Table 3.** Proximate analysis of native/torrefied sawdust.

| Specie | Type | Volatile Matter | Ash Content | Fixed Carbon |
|---|---|---|---|---|
| Larch sawdust | Native | 78.58 | 0.42 | 21 |
| | Torrefied | 61.53 | 0.47 | 38 |
| Oak sawdust | Native | 76.49 | 0.51 | 23 |
| | Torrefied | 53.42 | 0.58 | 46 |

This proximate analysis highlights that by torrefying the sawdust was enriched with fixed carbon, losing some of its volatile materials. Some differences were found regarding the fixed carbon content of torrefied sawdust, with the value of 21% higher for oak.

## 4. Discussion

The advantages of using the pelletizing process on larch sawdust have been highlighted by other researchers, namely the high durability [10,13] achieved by increasing the calorific value [1] by about 18% and by torrefying the sawdust followed by palletization [11,16]. Replacing nitrogen with rarefied air during torrefying is just one of the alternative methods that have been proposed, along with replacing it with saturated steam [15], with pressurized air [14] or with roasting gas [18]. All these methods aim to reduce the cost of sawdust torrefying, but the price of the torrefied pellets obtained using these methods does not yet justify their benefits.

The dimensions of wood particles are essential when it comes to obtaining pellets with superior characteristics, which is why the sawdust used in this study had an average size of 1.3 mm (Figure 1). In the same way, researchers [11,16] have established that the high properties of pellets are obtained from the small particles, and [18] has studied the influence of particle size on pellet characteristics. From the point of view of granulometry, maximum values of about 31% were obtained when 1.25 mm × 1.25 mm sieve was used, which means that the maximum dimensions of the chips were found in the area of the same sieve. Minimum values of about 2% were obtained when the 4 mm × 4 mm sieve was used, because the 5 mm × 5 mm sieve was used to remove the tendrils and also removed some of the large chips.

The three characteristics of the sawdust, namely bulk density, the expansion coefficient and the compression coefficient, were dependent on the dimensions of the chips with respect to their granulometry. Therefore, the values obtained were slightly different from those obtained by other authors [11].

Calorific values for wood pellets below 19,000 kJ/kg were also obtained by [3,12], a loss of mass below 34% by [18], and an ash content below 1% by [22]. The densities of solid wood were statistically analyzed (Table 4) by means of analysis of variance (ANOVA). The values of the F-value and *p*-value parameters highlight the normality of the value distribution for a 95% confidence interval, as other researchers have stated before [7,17].

**Table 4.** Analysis of variance for wood density.

| Source | DF | Adj SS | Adj MS | F-Value | *p*-Value |
|---|---|---|---|---|---|
| Larch density | 9 | 346.05 | 38.45 | 0.95 | 0.048 |
| Error | 1 | 40.50 | 40.50 | | |
| Total | 10 | 386.55 | | | |

In the same analysis, the pellets densities differed by less than 0.2% and the CV of the native pellets differed by less than 3.1%. Differences in the increase in calorific power after sawdust torrefaction were below 3.1% in favor of larch pellets [10]. Similar small differences were found for other features, such as caloric efficiency, energy release rate and calorific density [6].

The correlation between calorific value and degradation of main wooden compounds was achieved according to the main chemical compounds of wood, which are cellulose, hemi-cellulose and lignin. The influence of secondary chemicals (extractables and oxides from ash) was neglected. Of these major chemicals, lignin has the strongest influence on calorific value. Knowing that lignin has the highest calorific value, 25,121 kJ/kg (25.1 MJ/kg), while cellulose and hemicellulose have a value of about 17 374 kJ/kg (17.3 MJ/kg), an addictive relationship could be identified (17):

$$CV = 25{,}121 \times Li:100 + 17{,}374 \times (Ce + He):100 \; [kJ/kg] \tag{17}$$

where CV is the calorific value of sawdust in kJ/kg; Li is the content of lignin in % wt; Ce is the content of cellulose in % wt; and He is the content of hemicelluloses in % wt.

The average values for lignin, cellulose and hemicellulose content were taken from the literature [27], and are 32%, 46% and 22% for larch, respectively, and 32%, 45% and 23% for oak, respectively.

First, it should be considered that only hemicelluloses degrade up to 300 °C and cellulose is degraded after 300 °C. Due to the total degradation of hemicelluloses, only cellulose and lignin remained in the torrefied product, with proportions of 39.2% and 60.8%, respectively. Applying Equation (16) resulted in a calorific value of 20,354 kJ/kg for larch and 19,495 kJ/kg for oak.

Second, during the torrefying process, all the cellulose can be degraded with the hemicellulose, and the maximum calorific value is 25,121 kJ/kg for both species. In reality, after performing the torrefying test, the sawdust had a maximum calorific value of 21,040 kJ/kg in the case of the oak and 21,213 kJ/kg in the case of the larch, which means that only a certain amount of cellulose was degraded during the torrefaction process, about 6.5% for the larch and 13.7% for the oak. By looking closely at the torrefaction graphs (Figure 7), it can be found that the degradation of cellulose begins in the case of larch at a temperature of 220 °C and at 215 °C in the case of oak.In this time the cellulose degradation intensity wass lower than that of hemicelluloses.

## 5. Conclusions

Generally, wood biomass, such as larch and oak sawdust/pellets, is environmentally friendly and offers clean and quick energy without $CO_2$ emissions.

The native larch and oak pellets had few differences in terms of effective density and calorific value, but after the thermal treatment of torrefaction the calorific value grew significantly, up to 15.8%.

If a comparison of the calorific properties of larch and oak sawdust/pellets is made, it can be observed that, although they are two different species (softwood and hardwood) and have different densities (775 kg/m$^3$ in the case of oak and 521 kg/m$^3$ in the case of larch at 10% moisture content), their energetic properties after torrefaction are quite appropriate.

Considering that the two analyzed species (*Quercus robur* and *Larix decidua*) are valuable species which are mainly used in furniture, decorations, timber and veneers, etc., it is recommended that only their remnants from the processing of wood (sawdust, for example) should be used in the process of manufacturing native or torrefied pellets.

**Author Contributions:** Conceptualization, A.L.; methodology, C.S.; software, A.M.O.; validation, A.L.; formal analysis, A.M.O.; investigation, A.L.; resources, A.L.; data curation, C.S.; writing—original draft preparation, A.L.; writing—review and editing, A.L.; visualization, A.M.O.; supervision, A.M.O.; project administration, A.L. All authors have read and agreed to the published version of the manuscript.

**Funding:** This research received no external funding.

**Data Availability Statement:** Not applicable.

**Acknowledgments:** The authors would like to thank the Transilvania University of Brasov for their administrative and technical support.

**Conflicts of Interest:** The authors declare no conflict of interest.

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
