# Peer review of "Calorific Characteristics of Larch (Larix decidua) and Oak (Quercus robur) Pellets Realized from Native and Torrefied Sawdustâ€"

_forests, doi:10.3390/f13020361_

Round 1
Reviewer 1 Report
The article titled ‘Energetic aspects of oak and larch pellets obtained from torrefied sawdust waste’ is an interesting paper; however, some improvement need. Please see the below comments.
1. It will be better to include oak and larch scientific names in the title.
2. In the introduction, more references are needed about pellets.
3.The hypothesis of this article should be clearly described in the introduction section.
4. More exploration is needed in the discussion part, according to the results.

Author Response
Reviewer 1
We would like to thank the reviewer for his work and for his pertinent remarks. We believe that through the changes made, we have managed to meet the expectations of the reviewer and get a better work that can be closer to the FORESTS journal readers.
- It will be better to include oak and larch scientific names in the title.
Answer of authors: The scientific names were introduced in title.
- In the introduction, more references are needed about pellets.
Answer of authors: Another 3 recent papers in the field of pellets were introduced in the bibliography and were cited in the paper in the introduction area by adding a short paragraph.
- The hypothesis of this article should be clearly described in the introduction section.
Answer of authors: The objectives of the paper have been redone and simplified, in order to highlight the innovative elements of the research, namely the use of an oxygen-poor environment for treatment, the comparison between two structurally different species and the influence of moisture content on calorific value.
- More exploration is needed in the discussion part, according to the results.
Answer of authors: Three new paragraphs were introduced in the "Discussions" chapter, thus completing the discussions on other elements that had not been observed before.
Authors,
Reviewer 2 Report
The manuscript considers the energy characteristics of two types of lignocellulosic biomass. In general, the manuscript is written on a weak scientific and methodological level and does not provide new knowledge.
Point 1. The manuscript’s subject is similar to many works that study the properties of different types of biomasses, including oak and larch. Therefore, it is essential to indicate the novelty of the work.
Point 2. Proximate and ultimate analyses of biomasses should be provided. Their presence is necessary for the initial characterization of the energy potential of the biomass. The properties of the original biomass determine all further properties of the processed biomass, so it is unexpected why the authors did not provide them. This is the crucial point in the manuscript.
Point 3. In the title of the manuscript, you can indicate energy characteristics instead of energy aspects, which will be more consistent with the manuscript’s content.
Point 4. The introduction section is large and can be shortened without losing its meaning. Moreover, recent publications (last 2 years) should be added.
Point 5. Line 47. A reference at the end of the sentence is required ... will be soon finished.
Point 6. Requires a reference in caption Fig. 1, obtained based on known data.
Point 7. The references are not appropriately numbered because [10] and [11] are the same source.
Point 8. Line 90 and 94. The publication year of articles may be deleted.
Point 9. The list of references contains 28 sources, although according to the manuscript’s text, there are 37 sources.
Point 10. The description of the methods can be shortened, especially in cases where a well-known standard is used.
Point 11. Fig.2. The Figure on the right is unclear. A clear picture should be provided. Add a) and b) for Figures.
Point 12. Fig.3. The distribution of particles is given. If the authors would like to provide a granulometric distribution curve, so it is the total number of sizes. In addition, please, specify the percentage by mass or volume? As well as, it should be indicated below 0.4 instead of rest.
Point 13. Fig. 4. It is necessary to title two curves.
Point 14. Fig. 6, add a) and b) for Figures.
Point 15. Fig. 7, it is necessary to add a), b), and c) for the Figures. Additionally, arrange the Figures for 3, 5, and 10 min of torrefaction in sequence, and title the same type of Torrefaction temperature… or Temperature torrefaction… For the Figure for 3 min, it is necessary to correct Torefaction to Torrefaction. In addition, missing is the y-axis title.
Point 16. Fig. 8. Add a), b), and c) for Figures.
Point 17. Description of HCV change only for larch (Fig. 7) is redundant. The changes are clearly visible from the Figures, and there is no need to enumerate them without additional discussion.
Point 18. Line 475. The sentence should end with a reference.
Point 19. In addition to the ash content of the biomass, it is essential to know the composition of the ash. These results would be useful in the paper.
Point 20. Authors are advised to avoid "we" in the text of the article. And the paper should be subjected to proofreading and extensive grammar checking.
Author Response
We would like to thank the reviewer for his work and for his pertinent remarks. We believe that through the changes made, we have managed to meet the expectations of the reviewer and get a better work that can be closer to the journal readers.
Reviewer 2
- Point 1. The manuscript’s subject is similar to many works that study the properties of different types of biomasses, including oak and larch. Therefore, it is essential to indicate the novelty of the work.
Answer of authors: The objectives of the paper have been redone and simplified, in order to highlight the innovative elements of the research, namely the use of an oxygen-poor environment for thermal treatment, the comparison between two structurally different species and the influence of moisture content on calorific value, based on the original methods.
- Point 2. Proximate and ultimate analyses of biomasses should be provided. Their presence is necessary for the initial characterization of the energy potential of the biomass. The properties of the original biomass determine all further properties of the processed biomass, so it is unexpected why the authors did not provide them. This is the crucial point in the manuscript.
Answer of authors: - Regarding the chemical proximate analysis, out of the total tests of this analysis the next ones were performed: moisture content, ash content and calorific value. As only fixed carbon remained, a paragraph was introduced next to the results on ash content, showing the value of fixed carbon and volatile substances in the composition of sawdust-type biomass. In this way the information about this analysis is complete.
- Regarding the elementary chemical analysis (ultimate) we are working on a project to show the influence of the carbon content of different types of biomasses (including torrefied ones) on the calorific value. We do not have some conclusive results yet, which is why we do not dare to provide data on this analysis.
- We recognize the importance of these chemical analyses, but we have no results in this field, the purpose of our work being focused on increasing the calorific value.
- Point 3. In the title of the manuscript, you can indicate energy characteristics instead of energy aspects, which will be more consistent with the manuscript’s content.
Answer of authors: The term “characteristics” was introduced in the title.
- Point 4. The introduction section is large and can be shortened without losing its meaning. Moreover, recent publications (last 2 years) should be added.
Answer of authors: - Lots of paragraphs and phrases in the introductory part have been deleted. Also, 3 other new researches were introduced, corresponding to the last years (2021, 2020 and 2018). These new references have been properly cited in the paper, in the appropriate area of the introductory chapter.
- Point 5. Line 47. A reference at the end of the sentence is required ... will be soon finished.
Answer of authors: At the end of the sentence a new reference was introduced.
- Point 6. Requires a reference in caption Fig. 1, obtained based on known data.
Answer of authors: Reference [9] was introduced. This graph was made based on data collected from the literature.
- Point 7. The references are not appropriately numbered because [10] and [11] are the same source.
Answer of authors: Necessary changes were made where necessary.
- Point 8. Line 90 and 94. The publication year of articles may be deleted.
Answer of authors: Years of publication were deleted.
- Point 9. The list of references contains 28 sources, although according to the manuscript’s text, there are 37 sources.
Answer of authors: Reference in the text with “37” number was changed.
- Point 10. The description of the methods can be shortened, especially in cases where a well-known standard is used.
Answer of authors: The method descriptions were shorted by extraction of some paragraphs.
- Point 11. Fig.2. The Figure on the right is unclear. A clear picture should be provided. Add a) and b) for Figures.
Answer of authors: The requirements have been resolved, by clearing the right pictures and add “a” and “b” on the two pictures.
- Point 12. Fig.3. The distribution of particles is given. If the authors would like to provide a granulometric distribution curve, so it is the total number of sizes. In addition, please, specify the percentage by mass or volume? As well as, it should be indicated below 0.4 instead of rest.
Answer of authors: a.- We put “Below 0.4 mm” in figure. b. We specified “Mass percentage” on the Oy axis; c. Dimensional particle size (for length, width and thickness) is usually recommended for chipboard and OSB, where there are chips. This particle granulometry is not recommended for sawdust, where the 3 dimensions are almost similar.
- Point 13. Fig. 4. It is necessary to title two curves.
Answer of authors: We divided this figure in two graphs, noted with “a” and “b”.
- Point 14. Fig. 6, add a) and b) for Figures.
Answer of authors: The two figures 6 were denoted by “a” and “b”, in a single figure 6.
- Point 15. Fig. 7, it is necessary to add a), b), and c) for the Figures. Additionally, arrange the Figures for 3, 5, and 10 min of torrefaction in sequence, and title the same type of Torrefaction temperature… or Temperature torrefaction… For the Figure for 3 min, it is necessary to correct Torefaction to Torrefaction. In addition, missing is the y-axis title.
Answer of authors: - We added a, b and c part of Figure 7; - We arranged figures in normal order; We added an “r” in torrefaction word. -We put the terms on the Oy axis.
- Point 16. Fig. 8. Add a), b), and c) for Figures.
Answer of authors: The three parts of the figure were marked with a, b and c. The necessary addition was made in the legend of the respective figure.
- Point 17. Description of HCV change only for larch (Fig. 7) is redundant. The changes are clearly visible from the Figures, and there is no need to enumerate them without additional discussion.
Answer of authors: That description was erased. A general simpler statement about this paragraph was replaced instead.
- Point 18. Line 475. The sentence should end with a reference.
Answer of authors: A reference was introduced at the end of this paragraph.
- Point 19. In addition to the ash content of the biomass, it is essential to know the composition of the ash. These results would be useful in the paper.
Answer of authors: A new paragraph referring to this aspect was introduced in paper, in the sub-chapter of Ash content results.
- Point 20. Authors are advised to avoid "we" in the text of the article. And the paper should be subjected to proofreading and extensive grammar checking.
Answer of authors: We replaced all “we” from paper. Also, grammar checking of paper was made.
Round 2
Reviewer 2 Report
The authors have improved the article, but there are some points with which I am afraid I have to disagree.
Authors should show volatiles, ash, fixed carbon values for larch and oak sawdust in their initial state and after torrefaction in the Table. It will be more precise for readers. Presenting volatile and fixed carbon values within ranges is inappropriate; each parameter should have a specific value. Additionally, it is worth pointing out the equation the authors calculated fixed carbon. On line 626, the authors referred to [26]. Did the authors get permission to publish the data from [26]?
The captions for the axes in Figure 6 need to be standardized. The sizes of Figures 2a and 2b should be standardized. English still needs improvement.
Author Response
Reviewer The authors have improved the article, but there are some points with which I am afraid I have to disagree.
Authors should show volatiles, ash, fixed carbon values for larch and oak sawdust in their initial state and after torrefaction in the Table. It will be more precise for readers. Presenting volatile and fixed carbon values within ranges is inappropriate; each parameter should have a specific value. Additionally, it is worth pointing out the equation the authors calculated fixed carbon. On line 626, the authors referred to [26]. Did the authors get permission to publish the data from [26]?
The captions for the axes in Figure 6 need to be standardized. The sizes of Figures 2a and 2b should be standardized. English still needs improvement.
Author`s responses
In this way, the authors thank the activity submitted by the reviewer. We believe that this will make the paper more visible to future readers of the journal. Hire is our responses:
- In line with the requirements of the reviewer, a new Table 3 has been introduced at line 557, covering the values for volatile matter, ash content and fixed carbon, before and after torrefaction, for larch and oak sawdust.
- Discussions about the new features were introduced immediately after Table 3.
- Sub-chapter 3.11 has been changed, adding the new features introduced.
- As the new values were introduced, the corresponding citation was dropped.
- In the method chapter, the new thermal characteristics have been introduced accordingly.
- The dependency relationship between ash content, volatiles and fixed carbon has also been introduced in the method chapter, as Eq. 16.
- The two graphs in Figure 6 were joined, and the Ox axes were changed, moving the short form of the unit of measure 0C instead of the explicit form. Through these changes, it can consider that the captures in the axes of the figure have been standardized.
- The dimensions of the two images in Figure 2 were close, so that no differences could be seen between them.
- English was corrected by a professional expert.
Authors